# Antibiotic Susceptibility, Virulome, and Clinical Outcomes in European Infants with Bloodstream Infections Caused by Enterobacterales

**DOI:** 10.3390/antibiotics10060706

**Published:** 2021-06-11

**Authors:** Laura Folgori, Domenico Di Carlo, Francesco Comandatore, Aurora Piazza, Adam A. Witney, Ilia Bresesti, Yingfen Hsia, Kenneth Laing, Irene Monahan, Julia Bielicki, Alessandro Alvaro, Gian Vincenzo Zuccotti, Tim Planche, Paul T. Heath, Mike Sharland

**Affiliations:** 1Paediatric Infectious Disease Research Group, Institute for Infection and Immunity, St George’s University of London, Cranmer Terrace, London SW17 0RE, UK; yhsia@sgul.ac.uk (Y.H.); jbielick@sgul.ac.uk (J.B.); pheath@sgul.ac.uk (P.T.H.); msharland@sgul.ac.uk (M.S.); 2Department of Paediatrics, Vittore Buzzi Children Hospital, University of Milan, Via Lodovico Castelvetro 32, 20154 Milan, Italy; bresesti.ilia@asst-fbf-sacco.it (I.B.); gianvincenzo.zuccotti@unimi.it (G.V.Z.); 3Paediatric Clinical Research Centre “Romeo and Enrica Invernizzi”, Department of Biosciences, University of Milan, Via Giovanni Battista Grassi 74, 20157 Milan, Italy; domenico.dicarlo@unimi.it (D.D.C.); francesco.comandatore@unimi.it (F.C.); alessandro.alvaro@unimi.it (A.A.); 4Clinical-Surgical, Diagnostic and Pediatric Sciences Department, Unit of Microbiology and Clinical Microbiology, University of Pavia, Corso Str. Nuova 65, 27100 Pavia, Italy; aurora.piazza@unipv.it; 5Institute of Infection and Immunity, St George’s University of London, Cranmer Terrace, London SW17 0RE, UK; awitney@sgul.ac.uk (A.A.W.); klaing@sgul.ac.uk (K.L.); imonahan@sgul.ac.uk (I.M.); timothy.planche@stgeorges.nhs.uk (T.P.); 6School of Pharmacy, Queen’s University, 97 Lisburn Rd., Belfast BT9 7BL, UK; 7Pediatric Pharmacology and Pharmacometrics, University Children’s Hospital (UKBB), University Hospital Basel, Spitalstrasse 33, 4056 Basel, Switzerland

**Keywords:** infant, newborn, bacteremia, Gram-negative bacteria, drug resistance, microbial, virulence factors, mortality

## Abstract

Mortality in neonates with Gram-negative bloodstream infections has remained unacceptably high. Very few data are available on the impact of resistance profiles, virulence factors, appropriateness of empirical treatment and clinical characteristics on patients’ mortality. A survival analysis to investigate 28-day mortality probability and predictors was performed including (I) infants <90 days (II) with an available Enterobacterales blood isolate with (III) clinical, treatment and 28-day outcome data. Eighty-seven patients were included. Overall, 299 virulence genes were identified among all the pathogens. *Escherichia* *coli* had significantly more virulence genes identified compared with other species. A strong positive correlation between the number of resistance and virulence genes carried by each isolate was found. The cumulative probability of death obtained by the Kaplan-Meier survival analysis was 19.5%. In the descriptive analysis, early age at onset, gestational age at onset, culture positive for *E. coli* and number of classes of virulence genes carried by each isolate were significantly associated with mortality. By Cox multivariate regression, none of the investigated variables was significant. This pilot study has demonstrated the feasibility of investigating the association between neonatal sepsis mortality and the causative Enterobacterales isolates virulome. This relationship needs further exploration in larger studies, ideally including host immunopathological response, in order to develop a tailor-made therapeutic strategy.

## 1. Introduction

Mortality in neonates with Gram-negative bloodstream infections (GN-BSIs) has remained unacceptably high. Appropriate empirical treatment is considered crucially important in reducing mortality. However, despite the improvement in neonatal care, the fatality rate in babies with GN-BSIs remains around 15–20%, also during the emergence of antimicrobial resistance (AMR) [1,2,3,4,5,6].

Very few data are available on the impact of different treatment regimens on clinical outcome in neonates with GN-BSIs. Previous studies conducted in both adults and children showed conflicting results on the impact of resistance profiles, appropriateness of empirical treatment and clinical characteristics on patients’ mortality [7,8,9,10,11,12,13,14,15]. In the last decade, the implementation of modern bioinformatics to assist next-generation sequencing data analysis greatly improved the knowledge on the genetic characterization of pathogenic strains that may serve as target for new therapies [16]. There has been a growing body of evidence about the role of virulence factors (VFs) in the pathogenesis of invasive infections. Enterobacterales employ many strategies to enhance invasiveness, overcome host defenses and cause infections. Different strains can use alternative VFs with similar functions during the infection process, with this plasticity leading to unique combinations of such factors [17]. Some VFs are disease-specific whereas others seem to play different roles in different types of infection [18]. This plasticity enables pathogenic strains to colonize and infect different tissues and hosts. Some major classes of VFs, such as capsule, siderophores and fimbriae, have been characterized well [18]. However, several other factors were recently identified and have yet to be defined to fully understand their mechanisms of action and clinical significance (summarized in Table 1). To achieve this goal, a genomic approach can be used to identify genes encoding specific virulence determinants. In adults, several genes present in the great majority of bacteremic strains and involved in virulence have been identified [19,20,21]. These are presumably essential for the infection process. However, the specific VFs that are relevant in causing neonatal GN-BSIs are not well defined yet, partly due the variability among the few studies available so far [2]. These have mostly been conducted on neonates and children with *Escherichia coli* bacteremia, with virtually no data available on other Enterobacterales [22,23,24,25]. Also, in the recent years, several data have been published investigating a potential role of the bacterial virulome, defined as the set of genes contributing to the bacterial virulence, in determining the outcome of patients with both Gram-positive and GN-BSIs. Again, these studies demonstrate a mixed picture reporting a significant correlation between virulence factors and mortality in both children and adults [2,19,26,27,28,29].

Clarifying the role of the main determinants leading to adverse outcomes could help to define targeted interventions to decrease mortality. With this study, we aimed to investigate potential associations between patient characteristics, pathogen characteristics and antibiotic treatment regimen on the clinical outcome of neonates/infants affected by culture-proven GN-BSIs.

## 2. Results

### 2.1. Demographic and Clinical Data

Overall, 87 infants from six European countries (the United Kingdom: 49, Estonia: 21, Greece: 7, Italy: 7, Lithuania: 2, Spain: 1) between 2010–2015 fulfilled our inclusion criteria and were included in the study. Forty patients were retrieved from the neonatal infection surveillance network (NeonIN) study [30], 38 from NeoMero [31], and 9 from the Collaborations for Leadership in Applied Health Research and Care (CLAHRC) study [32]. Patients, pathogens and treatment characteristics of the included episodes are summarised in Table 2. At the BSI onset, the median age of the selected neonates was 15.2 days (interquartile range (IQR) 6.7–31), with a median gestational age (GA) of 33 weeks (IQR 28–37). Forty-nine out of 87 babies (56%) had a central line in situ at the episode onset. A total of 37 different antibiotic regimens have been reported among the 87 patients in the first 48 h of treatment.

### 2.2. Microbiological Data

The species IDs identified on matrix-assisted laser desorption/ionization time-of-flight (MALDI-TOF) mass spectrometry were all confirmed by sequencing. The isolate-specific accession numbers are indicated in Appendix A. The most frequently isolated pathogen was *E. coli* followed by *Enterobacter* spp. and *Klebsiella* spp. (Table 2). The percentage of multidrug-resistance (MDR) isolates was 30% (26/87). Based on interpretation of the in vitro susceptibility profile, 16/87 (18%) were suspected of producing an extended-spectrum beta lactamase (ESBL) enzyme, and only one isolate (*Klebsiella pneumoniae*) was resistant to carbapenems. Susceptibilities of single species to the investigated antibiotics are presented in Table 3.

A total of 50 different sequence types (STs) were found, with ST131 and ST90 as the most frequent in *E. coli* and *Enterobacter cloacae*, respectively. The median number of classes of resistance genes carried per isolate was 5 (IQR 4–5) whereas the median number of classes of virulence genes was 7 (IQR 7–9). Twenty-five isolates harbored blaTEM-type genes, two non-*Klebsiella* spp. strains the blaSHV-type determinant, and two *E. coli* strains the blaCTX-M-type genes. One *K. pneumoniae* ST17 carried blaVIM-12 gene, and two *Enterobacter asburiae* (ST484) the mcr-9 determinant.

Overall, the genome sequencing identified 299 different virulence genes among all the pathogens. There was a strong positive correlation between the number of resistance and virulence genes carried by each isolate (*Rho* = 0.79; *p* = 0.001) (Figure 1).

*E. coli* strains showed the highest mean number of virulence genes (105 vs. <65 in the other species overall), mainly those involved in fimbriae production (*p* < 0.0001) (Table 4). The genes that were more frequently carried by the isolates are summarized in Table 5. Among the most represented strains (*E. coli*, *E. cloacae*, *K. pneumoniae*, *K. oxytoca*) the following virulence genes were carried by all the isolates: *gal*, *gnd*, *rcs* (capsule); *ompA* (cell invasion); *ent*, *fep* (iron metabolism); *acr* (pumps). Some genes were shown to be strain-specific. Among them, the adhesion *mrk* gene cluster was sequenced in all *Klebsiella* spp. isolates as well as the secretion system’s *exe* and *impA.tss* genes. Conversely, genes encoding for motility and chemotaxis proteins (*che*, *flg* and *fli*) were only carried by *E. coli* and *Enterobacter* spp. The *clb*, *esp* and *hly* genes encoding for toxin proteins were sequenced only in *E. coli* strains. One hypervirulent *K. pneumoniae* with a hypermucoviscous phenotype harboring the *rmpA* and *wca* genes was found [33].

### 2.3. Determinants of 28-Day Case-Fatality

The cumulative probability of death obtained by the Kaplan-Meier survival analysis was 19.5% with the greater percentage of deaths happening in the first week. In the descriptive analysis, early age at onset (*p* = 0.002), culture positive for *E. coli* (*p* = 0.029), number of classes of virulence genes carried per isolate (*p* = 0.022) and GA (weeks) at the onset (*p* = 0.003) were significantly associated with mortality (Table 6). By Cox multivariate regression, none of the investigated variables was significant (Table 7).

## 3. Discussion

This study included 87 European neonates and infants younger than 90 days with GN-BSIs due to Enterobacterales. Overall, 299 virulence genes were identified in these pathogens. Among the different organisms, *E. coli* had significantly more virulence genes identified compared with other species. *Gal*, *gnd*, *rcs*, *omp*A, *ent*, *fep*, and *acr* virulence genes were identified from all the pathogens, likely being essential for the infection process. Conversely, some other genes were shown to be strain-specific. A strong positive correlation between the number of resistance and virulence genes carried by each isolate was found. By survival analysis, the 28-day probability of death was 19.5%. In the descriptive analysis, early age at onset, GA at the onset, culture positive for *E. coli* and number of classes of virulence genes carried by each isolate were significantly associated with mortality whereas discordant therapy was not related to mortality. By Cox multivariate regression, none of the investigated variables was significant.

Many studies have been conducted in both adults and neonates trying to define the main determinants of mortality in patients with GN-BSIs. AMR has been broadly investigated in the adult population, with the majority of studies reporting a significant association between multiple resistance to antibiotics and patients mortality [7,34,35,36]. Some large cohorts have not found a clear correlation between AMR and adverse outcome [8,12,13]. Data from this small study did not confirm a significant impact of resistance profile on neonatal mortality [5].

In recent years, an increasing number of studies are being conducted trying to investigate the impact of virulence genes on the outcome of patients with GN-BSIs. Among them, *E. coli* was the most frequently investigated pathogen followed by *K. pneumoniae*. Independent risk factors associated with 30-day mortality among adult patients with ESBL-producing *E. coli* bacteremia included siderophores *iro*N and *iss* positivity [21], the siderophore *fyu*A gene, and the presence of the afimbrial adhesin *afa* gene [19,37]. In a large prospective study investigating the main determinants for adverse outcome in patients with *K. pneumoniae* BSIs, the cytotoxicity *pks* gene cluster carriage by causative strains was an independent risk factor for 30-day mortality when accompanied by MDR [38]. Lastly, the siderophore-related *iut*A gene was found to be an independent predictor of the 30-day mortality in *K. pneumoniae* bacteremia [39]. However, almost all of these studies were conducted with pre-selected virulence genes searched by polymerase chain reaction (PCR) rather than sequencing the entire bacterial virulome. This led to a wide heterogeneity, with each group analyzing different genes and hampering the comparison of the results. Very little data have been published on the relationship between bacterial virulence factors and BSI mortality in children, and almost all in patients with *E. coli* bacteremia. In a prospective cohort of 43 septic neonates, the adhesin *hek*/*hra* gene was found to be significantly more frequent in isolates from newborns who died than in isolates from survivors [2]. On the other hand, in a cohort of children 0–17 years old (median age 2.4 months), none of the 20 virulence factors tested by PCR was found to correlate with sepsis severity [26]. The Burden of Antibiotic Resistance in Neonates from Developing Societies (BARNARDS) study was conducted to assess the burden of AMR in neonates in seven low-middle income countries [6]. In this study, Gram-negative (GN) pathogens from neonatal sepsis were isolated and characterized through whole genome sequencing (WGS) for resistance and virulence genes. The number of virulence genes carried by each isolate through a virulence score was used. The results obtained suggested that yersiniabactin and/or aerobactin/salmochelin virulence genes may be involved in a more rapid onset and mortality. However, the inability to follow up all neonates and additional local factors likely to contribute to patient’s death hampered the authors’ capacity to attribute mortality singularly to the presence/absence of genomic traits.

Our results showed a strong positive correlation between the number of resistance and virulence genes carried by each isolate. Many studies have been conducted on either AMR or virulence. However, the biological effect and connection between these two factors are of particular importance [40,41]. Indeed, a negative or positive relationship can be found among them. Enhanced virulence or AMR frequently has been reported to have a fitness cost on bacteria but their relationship changes according to different bacterial species, the resistance and virulence genes involved and the host’s immune system [42,43]. Some antibiotics, such as ceftazidime, cefotaxime and quinolones, have been reported to enhance the increase of the deletion and transposition of DNA regions that are specific for VFs [42]. By contrast, a positive correlation has been shown between AMR and virulence with the use of other antibiotics. In particular, uropathogenic strains of *E. coli* carrying the *bla*CTX-M-15 resistance gene also harbored more *col*V, *col*E2-E9, *col*Ia-Ib, *hly*A, and *csg*A genes as well as the *bla*OXA-2 beta lactamase was correlated with increased expression of *col*M, *col*B, *col*E, and *crl* genes [44]. Prophages are another mechanism that has been shown to be involved in both virulence and resistance diffusion through the spread of toxins to other resistant strains [45]. Porins and biofilm also play an important role in the relationship between virulence and resistance, with the first acting as a channel controlling the entrance of both antibiotics and VFs into the pathogens (i.e., the *Omp*C gene) and the second favoring antimicrobial treatment tolerance and infection persistence at the same time also increasing the transfer of resistance and virulence genes among the cells [46]. Lastly, an enhancement in both resistance and virulence characteristics of the pathogens can occur through mobile genetic elements like plasmids. These self-replicating extra-chromosomal elements are capable of transferring among different bacteria while carrying virulence and resistance genes [47]. This mechanism is independent of any antibiotic pressure.

There are several limitations in the study design, due to both confounding factors and heterogeneity of our sample. Firstly, the study is not powered to demonstrate a significant correlation between the presence of single resistance/virulence genes and neonates’ clinical outcome; this required us to analyze resistome and virulome by grouping genes according to the class. Moreover, we are well aware of the rapid dynamics of bacterial genetics, and the selection of a single time point isolate can cause bias and affect the results. At the same time, considering the possible implication of heteroresistance, the selection of certain colonies in the first place could have potentially missed relevant isolates. We found a significantly higher number of virulence genes in *E. coli* isolates compared with other species; this could be due to an over representation of *E. coli* genes in the virulence database (VFDB). Different pathogens can have different impacts on neonates with sepsis, and pooling data on multiple strains could alter the results. Lastly, neonates and infants represent a broadly heterogeneous population, being characterized by different gestational ages, birth weight, underlying conditions, and risk factors. Despite these limitations, this was the first study to correlate virulence factors of non-*E. coli* Enterobacterales with mortality in septic neonates. Although a number of studies have tried to correlate single virulence genes with Gram-negative sepsis outcome, virtually none of them have investigated the role of the entire virulome in causing mortality. Considering the huge number of virulence genes involved and the limited sample size, we presented a potential way to use the entire body of information gained from the WGS by grouping genes as the number of classes of virulence genes involved. Having access to a very complete dataset including patient-level treatment, outcome data, and isolates available for a comprehensive genetic characterization, we believe that our data provide new and relevant information on the molecular picture of GN pathogens causing neonatal BSIs.

## 4. Materials and Methods

### 4.1. Study Design and Data Source

A case series of neonates with clinical sepsis and microbiologically confirmed GN-BSIs was constructed from three separate studies: the NeonIN [30], the NeoMero clinical trial [31], and the CLAHRC [32]. The NeonIN is a multinational network which prospectively collect data on neonatal infections from neonatal units. The detailed data procedures have been previously described [48]. NeoMero was a European-based randomized controlled trial to compare the efficacy of meropenem with standard of care (ampicillin + gentamicin or cefotaxime + gentamicin) in the treatment of late onset neonatal sepsis. CLAHRC is a United Kingdom-based prospective cohort study to collect data for patients with GN-BSIs in three hospitals in South London which aims to characterize clinical management of patients with GN-BSIs in all ages and identify potential risk factors associated with 28-day treatment outcomes.

### 4.2. Selection Criteria, Available Data and Definitions

Patients with a microbiologically-confirmed diagnosis of GN-BSIs due to Enterobacterales were selected from the above studies among European Neonatal Intensive Care Units between 2010–2015. Inclusion criteria were (I) age between 0–90 days, (II) Enterobacterales blood isolate available for the sequencing, (III) patient-level clinical and treatment data, and (IV) 28-day clinical outcome data.

Data available for the analysis included demographic, risk factors, clinical characteristics, antibiotic treatment, susceptibility results, resistome and virulome of the isolated bacteria, and 28-day treatment outcome.

The date of the blood culture was considered as the day of the sepsis onset. The MDR of the isolates was defined according to Magiorakos A.-P. et al. as acquired non-susceptibility to at least one agent in three or more antimicrobial categories [49]. The first 48-h antibiotic treatment was defined as concordant or discordant based on the inclusion of at least an active drug against the blood isolate. The evaluated outcome was defined as 28-day case-fatality (patient alive/dead).

### 4.3. Microbiological Methods

Bacterial isolates from blood were cultured from frozen stocks (−80 °C) on blood agar plates. Species were identified by MALDI-TOF mass spectrometry (Bruker, Karlsruhe, Germany). Antibiotic susceptibility profiles were obtained according to the European Committee on Antimicrobial Susceptibility Testing (EUCAST) 2019 Clinical Breakpoints with disk diffusion tests for the following antibiotics: amikacin, ampicillin, amoxicillin-clavulanate, ceftriaxone, ceftazidime, aztreonam, ciprofloxacin, gentamicin, meropenem, piperacillin-tazobactam, and trimethoprim-sulphametoxazole. To facilitate the analyses, isolates that were defined as having increased exposure were classified as non-susceptible. The isolates included in the study were subjected to WGS using the Illumina MiSeq platform (Illumina, San Diego, CA, USA), with paired-end runs of 2 × 300 bp, after Nextera XT library preparation. The obtained reads were assembled using SPAdes [50]. For each genome, we determined the ST using an in-house script (available upon request) and the Multilocus sequence typing (MLST) schemes and gene alleles sequences available on PubMLST (pubmlst.org). The isolates were further characterized at the genomic level with the identification of resistant and virulence genes using ABRicate (Seemann T, Abricate, Github https://github.com/tseemann/abricate, accessed on: 11 November 2019) and the following databases: The Comprehensive Antibiotic Resistance Database (CARD) [51] and Resfinder [52] for the resistance genes and VFDB [53] for the virulence genes.

### 4.4. Statistical Analysis

Qualitative variables were summarized by absolute frequencies and percentages, and quantitative variables by median and IQR. A descriptive analysis was conducted with the potential association between variables and outcome of interest evaluated by chi-squared or Fisher’s exact test as more appropriate for qualitative variables, and Mann–Whitney or t-test as appropriate for quantitative variables. A Spearman’s rank correlation was calculated to evaluate the correlation between the number of resistance genes and virulence genes carried by each isolate. We performed a survival analysis to investigate the 28-day mortality predictors using the Cox regression model (primary endpoint), after evaluated the proportional hazard (PH) assumption. In case of non-PH assumption, the weighted Cox regression model was performed [54]. A bivariate analysis (univariate) was carried out and the variables for which the *p*-value was <0.10 in univariate analysis were included in the multivariate model. All variables entered as covariates were evaluated at the baseline. As secondary endpoint, a survival analysis was conducted using the Kaplan-Meier curves to assess the probability of death at 28 days. To facilitate the analysis, the resistome and virulome data obtained in sequencing were categorized as number of classes of virulence and resistance genes carried by each isolate. Because of the huge heterogeneity of treatment regimens among the included patients, antibiotics were coded according to the WHO ATC/DDD Index 2020 at the 4th level [55]. *p* values <0.05 were considered as statistically significant.

All statistical analyses were performed using R Statistical Software (version 4.0.2) [56].

### 4.5. Ethics

The source studies were approved by the Ethical Committees of the participating institutions, and all enrolled patients’ legal guardians provided informed consent. Given the retrospective nature of the present study, ethical approval for this analysis was not necessary.

## 5. Conclusions

To conclude, this pilot study demonstrated the feasibility of investigating the association between neonatal sepsis mortality and the causative Enterobacterales isolates virulome. The limited sample size of our cohort did not allow us to determine the role of single virulence genes in neonatal GN-BSIs but grouping genes as the number of classes involved allowed us to investigate the impact of the entire virulome in neonatal sepsis outcome. This knowledge may be useful for predicting clinical outcomes, detecting virulent strains, and helping with vaccine development. Expanding research on anti-virulence molecules together with the development of new antibiotics could be crucial to improving the management of these fragile patients. Further research would be advisable to elucidate the correlation with the timing of sepsis onset to personalize the clinical approach. These findings need further exploration in larger global studies, ideally including host immunopathological response, in order to develop a tailor-made therapeutic strategy.

## Figures and Tables

**Figure 1 antibiotics-10-00706-f001:**
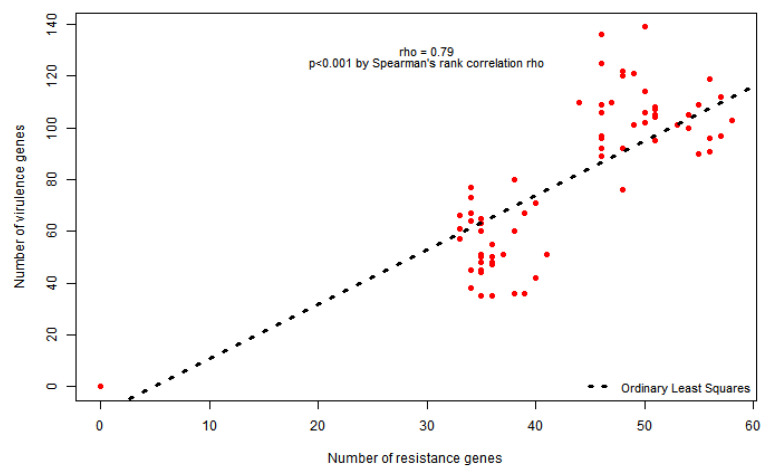
Spearman’s rank correlation between number of resistance and virulence genes.

**Table 1 antibiotics-10-00706-t001:** Main virulence factors in Enterobacterales.

Category	Sub-Category	Function	Genes	Pathogens
Adherence	Anti-aggregation protein-dispersin	Bound to the outer membrane, assisting dispersion across the surface by overcoming electrostatic attraction between fimbriae and bacterial surface	*aap*	*Escherichia coli*
	Adhesins	Cell-surface components that allow bacteria to attach to host cells or to surfaces	*afa*, *dra*, *fde*	*Escherichia coli*
	Fimbriae	Major adhesive structures in biofilm formation and binding to abiotic surfaces	*agg*, *bcf*, *csg*, *daa*, *fim*, *foc*, *lpf*, *mrk*, *paa*, *pap*, *sfa*, *yag.ecp*, *ykg.ecp*	*Escherichia coli* *Enterobacter* spp. *Klebsiella* spp.
	Intimin	Outer membrane protein needed for intimate adherence	*eae*, *tir*	*Escherichia coli*
	Zinc metalloprotease	Contributes to intimate adherence to host cells	*stc*	*Escherichia coli*
Bacterial metabolism	Allantoin Metabolism	Enzymes involved in degradation of allantoin	*all* A-D	*Klebsiella* spp. *Escherichia coli* *Enterobacter* spp.
	Transcription factors	DNA-binding transcriptional activator/repressor	*all* R-S	*Klebsiella* spp. *Escherichia coli* *Enterobacter* spp.
	Bacterial survival promoters	Methionine aminopeptidase	*map*	*Escherichia coli*
		Magnesium transporter	*mgt*	*Klebsiella* spp. *Escherichia coli* *Enterobacter* spp.
		Toll-like receptor and MyD88-specific signalling inhibitor	*tcp*	*Escherichia coli*
Capsule	Capsule	Extracellular polysaccharide matrix that envelops the bacteria, prevents phagocytosis, hinders the bactericidal action of antimicrobial peptides, blocks complement components	*cps*, *gal*, *glf*, *gnd*, *gtr*, *kfo*, *kps*, *man*, *rcs*, *rmp*, *ugd*, *wca*, *wza*, *wzi*, *wzm*, *wzt*	*Klebsiella* spp. *Escherichia coli* *Enterobacter* spp.
	Lipopolysaccharide	Component of the outer leaflet of the cell membrane of all Gram-negative bacteria (GNB) which protects against humoral defences	*lpx*, *waa*, *wbb*	All GNB
Cell invasion	Arylsulfatase	Penetration of the blood-brain barrier	*asl*	*Escherichia coli*
	Outer membrane porin A	Adherence to epithelial cells, translocation into epithelial cells nucleus, induction of epithelial cell death, biofilm formation, binding to factor H to allow bacteria to develop serum-resistance	*omp*A	*Escherichia coli* *Enterobacter* spp. *Klebsiella* spp.
	Invasion protein A	Cell invasion into the host tissues	*ibe*A	*Escherichia coli*
Iron metabolism	Siderophores-Hemin uptake	Enable using of Fe from haemoglobin in the host system	*chu*	*Escherichia coli* *Enterobacter* spp.
	Siderophores-Enterobactin	Mediation of iron acquisition, obstacole macrophages antimicrobial responses	*ent*, *fep*, *fes*	All GNB
	Siderophores-Yersiniabactin	Can solubilise iron bound to host binding proteins and transport it back to the bacteria	*fyu*, *irp*, *ybt*	*Escherichia coli* *Klebsiella* spp.
	Siderophores-Salmochelin	Siderophore receptor, use of Fe ions obtained from the body host	*iro*	*Escherichia coli* *Enterobacter* spp. *Klebsiella* spp.
	Siderophores-Aerobactin	Acquisition of Fe^2+/3+^ in the host system	*iuc*, *iut*	*Escherichia coli* *Klebsiella pneumoniae*
	Heme/haemoglobin transport protein and receptor	Cell survival	*shu*	*Escherichia coli* *Enterobacter* spp.
Motility and chemotaxis	Chemotaxis	Bacterial movement in response to a chemical stimulus	*che*, *mot*	*Escherichia coli* *Enterobacter* spp.
	Flagella	Motility organelle, function as adhesins	*flg*, *flh*, *fli*	*Escherichia coli* *Enterobacter* spp.
Pumps	Pumps	Efflux pump implicated in both virulence and resistance to antibiotics	*acr*	*Escherichia coli* *Enterobacter* spp. *Klebsiella* spp.
Secretion system factor	Type I secretion system protein (T1SS)	Enables pathogens to inject effector proteins into host cells	*hly*	*Escherichia coli*
	Type II secretion system protein (T2SS)	Enables pathogens to inject effector proteins into host cells	*exe*, *gsp*	*Escherichia coli* *Klebsiella* spp.
	Type III secretion system (T3SS)	Enables pathogens to inject effector proteins into host cells	*ces*, *esc*	*Escherichia coli*
	Type VI secretion system (T6SS)	Enables pathogens to inject effector proteins into host cells	*clp*V*.tss*H, *dot*U*.tss*L, *hcp*, *hsi*B1*.vip*, *icm*F*.tss*	*Escherichia coli* *Enterobacter* spp. *Klebsiella* spp.
Toxins	Colibactin	Genotoxin causing genomic instability in eukaryotic cells by induction of double-strand breaks in DNA	*clb*	*Escherichia coli* *Klebsiella pneumoniae*
	T3SS effector	Cytoskeletal rearrangements	*esp*	*Escherichia coli*
	Hemolysin A	Creating of pores in membranes of host cells (cell lysis)	*hly*	*Escherichia coli*

**Table 2 antibiotics-10-00706-t002:** Demographic and clinical characteristics of included patients.

Variable	Overall, *n* = 87 (%)
Gender	
Male	43 (49)
Female	44 (51)
Age at the onset (days, median (IQR))	15.2 (6.7–31)
Gestational age category (weeks of GA)	
<28 0/7	37 (42)
28 0/7–31 6/7	21 (24)
32 0/7–33 6/7	8 (9)
34 0/7–36 6/7	8 (9)
37 0/7–38 6/7	7 (8)
39 0/7–40 6/7	6 (7)
Birth weight category (grams)	
>=2500	20 (23)
1500–<2500	15 (17)
1000–<1500	15 (17)
<1000	37 (42)
Small for Gestational Age (SGA)	
Yes	11 (13)
No	76 (87)
Underlying conditions	
Yes	51 (59)
No	36 (41)
Gestational age (weeks) at onset, median (IQR)	33 (28–37)
Isolated organism	
*Escherichia coli*	36 (41)
*Enterobacter cloacae*	18 (21)
*Klebsiella pneumoniae*	11 (13)
*Klebsiella oxytoca*	7 (8)
*Serratia marcescens*	7 (8)
*Enterobacter asburiae*	3 (3)
*Enterobacter aerogenes*	2 (2)
*Serratia liquefaciens*	1 (1)
*Enterobacter kobei*	1 (1)
*Proteus mirabilis*	1 (1)
First 48-h antibiotic treatment *	
Aminoglycosides anti-bacterials	25 (29)
Beta-lactam anti-bacterials, penicillins	26 (30)
Other anti-bacterials	9 (10)
Other beta-lactam anti-bacterials	23 (26)
Quinolone anti-bacterials	4 (5)
First 48-h treatment concordance with the anti-biogram	
Concordant	81 (93)
Discordant	6 (7)
Multidrug resistant **	
No	61 (70)
Yes	26 (30)
Number of classes of resistance genes per isolate, median (IQR)	5 (4–5)
Number of classes of virulence genes per isolate, median (IQR)	7 (7–9)

* Coded according to the WHO ATC/DDD (Anatomical Therapeutic Chemical/Defined Daily Dose) Index 2020 at the 4th level. ** According to Magiorakos, A.P., *Clin. Microbiol. Infect.* **2012** Mar, *18*(3), 268–281.

**Table 3 antibiotics-10-00706-t003:** Percentages of susceptibility for the isolated pathogens.

Pathogen (*n*)	AMK *	AMP	AMC	CRO	CAZ	ATM	CIP	GEN	MEM	TZP	SXT
*Escherichia coli* (36)	97	42	67	94	97	94	89	86	100	92	64
*Enterobacter cloacae* (18)	94	0	0	56	56	83	100	67	100	94	89
*Klebsiella pneumoniae* (11)	55	0	46	46	55	64	91	64	91	64	64
*Klebsiella oxytoca* (7)	100	0	71	86	86	100	100	71	100	100	86
*Serratia marcescens* (7)	100	0	0	86	100	86	86	100	100	86	100
*Enterobacter asburiae* (3)	100	0	0	100	100	100	100	33	100	100	100
*Enterobacter aerogenes* (2)	50	0	0	50	50	50	100	50	100	50	100
*Serratia liquefaciens* (1)	100	100	100	100	100	100	100	100	100	100	100
*Enterobacter kobei* (1)	100	100	0	100	100	100	100	100	100	100	100
*Proteus mirabilis* (1)	100	0	100	100	100	100	100	0	100	100	0

AMK: amikacin, AMP: ampicillin, AMC: amoxicillin-clavulanate, CRO: ceftriaxone, CAZ: ceftazidime, ATM: aztreonam, CIP: ciprofloxacin, GEN: gentamicin, MEM: meropenem, TZP: piperacillin-tazobactam, SXT: trimethoprim-sulphametoxazole * Proportion of isolates resistant to the antibiotic.

**Table 4 antibiotics-10-00706-t004:** Median number of classes of virulence genes carried per bacterial strain.

Pathogen (*n*)	Virulence Factors Category (Median Number of Genes Carried)								
	Adherence	Bacterial Metabolism	Capsule	Cell Invasion	Iron Metabolism	Motility and Chemotaxis	Pumps	Secretion System Factor	Toxins
*Escherichia coli* (36)	31	2	5.5	2	37	8	2	13	6.5
*Enterobacter cloacae* (18)	5	0	7	1	16	7	2	11	0
*Klebsiella pneumoniae* (11)	24	0	14	1	12	0	2	13	0
*Klebsiella oxytoca* (7)	15	2	9	1	22	0	2	10	0

**Table 5 antibiotics-10-00706-t005:** Number and percentage of the most represented virulence genes.

Virulence Gene	*Escherichia coli* (36)	*Enterobacter cloacae* (18)	*Klebsiella pneumoniae* (11)	*Klebsiella oxytoca* (7)
	*N* (%)	*N* (%)	*N* (%)	*N* (%)
**Adherence**				
*csg*	36 (100)	18 (100)	0	0
*fde*	36 (100)	0	0	0
*fim*	36 (100)	0	10 (91)	7 (100)
*mrk*	0	2 (11)	11 (100)	7 (100)
*pap*	26 (72)	0	0	0
*yag.ecp*	36 (100)	0	11 (100)	7 (100)
*ykgK.ecp*	35 (97)	0	11 (100)	0
**Bacterial metabolism**				
*all*	36 (100)	0	0	3 (43)
*mgt*	0	0	0	7 (100)
**Capsule**				
*cps*	0	0	10 (91)	0
*gal*	36 (100)	18 (100)	11 (100)	7 (100)
*gif*	0	0	6 (54)	7 (100)
*gnd*	36 (100)	18 (100)	11 (100)	7 (100)
*kps*	33 (92)	0	0	0
*man*	0	18 (100)	7 (64)	2 (29)
*rcs*	36 (100)	18 (100)	11 (100)	7 (100)
*ugd*	2 (6)	18 (100)	11 (100)	7 (100)
**Cell invasion**				
*asl*	36 (100)	0	0	1 (14)
*ompA*	36 (100)	18 (100)	11 (100)	7 (100)
**Iron metabolism**				
*chu*	35 (97)	6 (33)	0	0
*ent*	36 (100)	18 (100)	11 (100)	7 (100)
*fep*	36 (100)	18 (100)	11 (100)	7 (100)
*fes*	36 (100)	0	11 (100)	7 (100)
*fyu*	35 (97)	0	2 (18)	5 (71)
*iro*	11 (31)	11 (61)	11 (100)	0
*irp*	35 (97)	0	2 (18)	5 (71)
*iuc*	25 (69)	0	1 (9)	0
*iut*	25 (69)	0	1 (9)	0
**Motility and chemotaxis**				
*che*	36 (100)	18 (100)	0	0
*flg*	36 (100)	18 (100)	0	0
*fli*	36 (100)	18 (100)	0	0
Pumps				
*acr*	36 (100)	18 (100)	11 (100)	7 (100)
**Secretion system factor**				
*clpV.tssH*	0	13 (72)	11 (100)	3 (43)
*dotU.tssL*	0	13 (72)	11 (100)	4 (57)
*exe*	0	0	11 (100)	7 (100)
*gsp*	35 (97)	0	0	0
*hcp*	21 (58)	14 (78)	11 (100)	7 (100)
*hsiB1.vip*	0	18 (100)	0	0
*icmF.tss*	0	11 (61)	10 (91)	4 (57)
*impA.tss*	0	0	11 (100)	0
*vasE.tssK*	0	12 (67)	11 (100)	4 (57)
*vip.tss*	21 (58)	15 (83)	11 (100)	7 (100)
*ybd*	0	18 (100)	10 (91)	7 (100)
**Toxins**				
*clb*	7 (19)	0	1 (9)	0
*esp*	15 (42)	0	0	0

**Table 6 antibiotics-10-00706-t006:** Descriptive analysis of potential association between variables and mortality.

Variable	Alive, *n* = 69 (%)	Died, *n* = 18 (%)	*p*-Value
Male	31 (45)	12 (67)	0.168
Female	38 (55)	6 (33)	
Age at the onset (days, median (IQR))	19.1 (8.8–35)	7.1 (3.3–9.8)	**0.002**
Gestational age category (weeks of GA)			
<28 0/7	27 (39)	10 (56)	0.323
28 0/7–31 6/7	17 (25)	4 (22)	1.000
32 0/7–33 6/7	7 (10)	1 (6)	1.000
34 0/7–36 6/7	8 (12)	0 (0)	0.197
37 0/7–38 6/7	5 (7)	2 (11)	0.631
39 0/7–40 6/7	5 (7)	1 (6)	1.000
Birth weight category (grams)			
>=2500	17 (25)	3 (17)	0.754
1500–<2500	13 (19)	2 (11)	0.727
1000–<1500	13 (19)	2 (11)	0.727
<1000	26 (38)	11 (61)	0.128
Small for Gestational Age (SGA)			
Yes	8 (12)	3 (17)	0.690
No	61 (88)	15 (83)	
Underlying conditions			
Yes	39 (56)	12 (67)	0.610
No	30 (43)	6 (33)	
Gestational age (weeks) at onset, median (IQR)	33 (30–38)	27 (25–33)	**0.003**
Isolated organism			
*Escherichia coli*	24 (35)	12 (67)	**0.029**
*Enterobacter* spp.	20 (29)	4 (22)	0.769
*Klebsiella* spp.	16 (23)	2 (11)	0.342
*Serratia* spp/*Proteus mirabilis*	9 (13)	0 (0)	0.194
First 48-h antibiotic treatment *			
Aminoglycosides antibacterials	19 (27)	6 (33)	0.848
Beta-lactam antibacterials, penicillins	22 (32)	4 (22)	0.611
Other antibacterials	7 (10)	2 (11)	1.000
Other beta-lactam antibacterials	19 (27)	4 (22)	0.770
Quinolone antibacterials	2 (3)	2 (11)	0.188
First 48-h treatment concordance with the antibiogram			
Concordant	64 (93)	17 (94)	1.000
Discordant	5 (7)	1 (6)	1.000
Multidrug resistant			
No	47 (68)	14 (78)	1.000
Yes	22 (32)	4 (22)	1.000
Number of classes of resistance genes per isolate, median (IQR)	5 (4–5)	5 (4–5)	0.203
Number of classes of virulence genes per isolate, median (IQR)	7 (7–9)	9 (8–9)	**0.022**

* coded according to the WHO ATC/DDD Index 2020 at the 4th level.

**Table 7 antibiotics-10-00706-t007:** Multivariate regression analysis on the 28-day mortality predictors.

Variable	N	HR * (L.95–U.95)	*p*-Value
Age at the onset (days)	87	0.97 (0.93–1.01)	0.125
N. classes of virulence genes	87	1.35 (0.92–1.97)	0.128
Gestational Age (weeks) at onset	87	0.91 (0.83–1)	0.056

* Hazard Ratio.

## Data Availability

The corresponding author confirms that he had full access to all the data in the study and had final responsibility for the decision to submit for publication. Whole-genome shotgun projects have been deposited in GenBank (project code PRJEB44870). The isolate-specific accession numbers are indicated in Appendix A.

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
