# Peer review of "Antibiotic Susceptibility, Virulome, and Clinical Outcomes in European Infants with Bloodstream Infections Caused by Enterobacterales"

_antibiotics, 2021, doi:10.3390/antibiotics10060706_

Round 1

Reviewer 1 Report

The present study examined the relationship between pathogen characteristics, antibiotic treatment choices and clinical outcome in neonates/infants with confirmed Gram-negative blood stream infections. The manuscript is well-written in clear English and provides important information about characteristics of infection and clinical outcome predictors in neonatal/infantile GN-BSI. The study design is appropriate and well-thought out. I think the paper would benefit from expanding the conclusions a little or explaining in a sentence or two how this pilot study specifically demonstrated feasibility in investigating the association between neonatal sepsis mortality and the infecting organism's virulome. 

Author Response

1- I think the paper would benefit from expanding the conclusions a little or explaining in a sentence or two how this pilot study specifically demonstrated feasibility in investigating the association between neonatal sepsis mortality and the infecting organism's virulome.

1- Thank you for this comment. We added few sentences at the end of the discussion and expanded the conclusion as requested.

Reviewer 2 Report

The authors describe characteristics of gram-negative bacterial isolates from 87 neonates with bacteremia. The study is well designed and clear in methodology. I believe that there are too many tables, graphics which make the paper difficult to read due to the shear number of data points. 

Comments:

  • 2. Results: the text in this section should be substantially reduced as their is significant duplication between text and table. If the data are available in the table, feel free to send the reader to the table - please do not duplicate the values in text; it makes the paper difficult to read.
  • 2.2 also has significant dupliacation (see previous comment)
  • Table 3: typically, data tables such as these in other studies list the % susceptible instead of the % resistant. Antibiograms also list % susceptible. The table would be much easier to read if the % were switched to % susceptible
  • Figure 1: can the authors supply the r2 (r-squared) value instead of rho (or in addition to rho)
  •  Figure 1: I recommend removing the isolate with 0 resistance and virulence genes. It is significantly affecting the spearman rank correlation and should be considered an outlier. This test should be performed without that isolate
  • Table 4: I'm not sure if tables 4 and 5 add much to the paper. Could these be summarize briefly in text instead?
  • Figure 2: as there is no comparator group, not sure that figure 2 is necessary. The curve is exactly what we would predict, greater percentage of deaths in the first week followed by slow increase in mortality. I recommend removing the graph. 

Author Response

2- Results: the text in this section should be substantially reduced as there is significant duplication between text and table. If the data are available in the table, feel free to send the reader to the table - please do not duplicate the values in text; it makes the paper difficult to read.

2- We agree with your comment. The results section has been shortened and the duplication between text and table removed.

3- Table 3: typically, data tables such as these in other studies list the % susceptible instead of the % resistant. Antibiograms also list % susceptible. The table would be much easier to read if the % were switched to % susceptible

3- Table 3 has been changed as suggested.

4- Figure 1: can the authors supply the r2 (r-squared) value instead of rho (or in addition to rho)

4- Thank you for asking this question. The R-squared index is typical of the Pearson correlation test which can be applied when 1) data have a normal distribution; 2) it is possible to hypothesize a linear correlation. Since our data have not a normal distribution, we applied the Spearman test which does not return the r-squared index but only rho.

5- Figure 1: I recommend removing the isolate with 0 resistance and virulence genes. It is significantly affecting the spearman rank correlation and should be considered an outlier. This test should be performed without that isolate

5- Thank you for this consideration. However, restricting the analysis to a selected population would expose to a colliding bias and can induce associations that are not real.

6- Table 4: I'm not sure if tables 4 and 5 add much to the paper. Could these be summarise briefly in text instead?

6- The virulence genes/mechanisms are the focus of this paper as well as the Special Issue itself. These tables were the best way we found to present the greater amount of information about the virulome of the isolates we investigated. It would be very difficult to summarise them in a plain text as it would end up as a list of genes and percentages and it wouldn’t allow us to highlight the differences between the various pathogens.

7- Figure 2: as there is no comparator group, not sure that figure 2 is necessary. The curve is exactly what we would predict, greater percentage of deaths in the first week followed by slow increase in mortality. I recommend removing the graph.

7- The graph has been removed as you suggested.